# The “Little Iron Waltz”: The Ternary Response of *Paracoccidioides* spp. to Iron Deprivation

**DOI:** 10.3390/jof6040221

**Published:** 2020-10-12

**Authors:** Aparecido Ferreira de Souza, Marcella Silva de Paula, Raisa Melo Lima, Marielle Garcia Silva, Juliana Santana de Curcio, Maristela Pereira, Célia Maria de Almeida Soares

**Affiliations:** Laboratório de Biologia Molecular, Instituto de Ciências Biológicas, ICB II, Campus II, Universidade Federal de Goiás, Goiânia 74000-000, Brazil; aparecidofsouza@gmail.com (A.F.d.S.); mar_cella10sp@hotmail.com (M.S.d.P.); raisamelolima@hotmail.com (R.M.L.); mariellegarcias@gmail.com (M.G.S.); julianadecurcio1@gmail.com (J.S.d.C.); maristelaufg@gmail.com (M.P.)

**Keywords:** siderophores, CFEM proteins, reductive iron uptake pathway

## Abstract

*Paracoccidioides* is a genus of thermodimorphic fungi that causes paracoccidioidomycosis. When in the host, the fungus undergoes several challenges, including iron deprivation imposed by nutritional immunity. In response to the iron deprivation triggered by the host, the fungus responds in a ternary manner using mechanisms of high affinity and specificity for the uptake of Fe, namely non-classical reductive iron uptake pathway, uptake of host iron proteins, and biosynthesis and uptake of siderophores. This triple response resembles the rhythmic structure of a waltz, which features three beats per compass. Using this connotation, we have constructed this review summarizing relevant findings in this area of study and pointing out new discoveries and perspectives that may contribute to the expansion of this “little iron waltz”.

## 1. Introduction

One of the greatest things about studying the host–pathogen interaction is to gradually understand that despite the storm of molecular events that support the adaptation process, there is organization! This organization is often referred to as “orchestration” or an “orchestrated response” [1,2,3]. It truly makes sense to trace an association between molecular events and the performance of music in an orchestrated way; in both cases, there is a myriad of individuals acting, albeit in an organized manner, which provides a final result of completeness. The host, through its immunity, employs several strategies that generate stress to pathogens, which in most cases makes it impossible for microorganisms to persist in it [4,5]. One of these strategies is nutritional immunity, a network of processes that culminate in limiting the bioavailability of nutrients and micronutrients for pathogens [6,7,8]. Without nutrients, there are two alternatives for microorganisms: succumbing to death or adapting to an environment of scarcity [9].

Iron is one of the micronutrients that is deprived through nutritional immunity. This transition metal is widely employed as a cofactor of several enzymes involved in crucial cellular processes, such as DNA synthesis and repair, energy metabolism, antioxidant systems, and biopolymer synthesis, among many others [10]. Obviously, iron is important for both the host and the pathogen, given its participation in elementary processes for life. This massive recruitment of iron is due to its flexibility of oxidation states, switching from ferric ion (Fe^3+^) to ferrous ion (Fe^2+^) with remarked facility [11]. Like a double-edged sword, this transition metal characteristic also makes it harmful if free in the intracellular environment, as this electron movement can damage biomolecules, which require that both host and pathogens employ mechanisms for the strict control of metal’s bioavailability [12,13].

In the context of the host–pathogen interface, it is advantageous for the host, through nutritional immunity, to limit the bioavailability of iron to pathogens, given the metal’s importance for the maintenance of vital cellular processes; if pathogens do not have access to the metal, the inevitable path is their death [14]. However, pathogens that have infectious success, like pathogenic bacteria and protozoa, are able to supplant nutritional immunity through highly sensitive and specific mechanisms for iron uptake [9,15]. Regarding pathogenic fungi, this is also a reality. In a generalist way, these microorganisms respond to iron deprivation by three main events: (1) the reductive iron assimilation pathway (as described for *Cryptococcus neoformans*); (2) exploration of iron-containing host proteins, such as hemoglobin (as described for *Candida albicans*) and; (3) biosynthesis and uptake of siderophores (as described for *Aspergillus fumigatus*) [16,17].

Latin America is the endemic region of a genus of pathogenic fungi named *Paracoccidioides*, which causes paracoccidioidomycosis (PCM), a systemic mycosis that affects both immunocompetent and immunocompromised individuals and can cause severe sequelae and even death, if proper treatment is not employed [18,19]. Despite the extensive work found in the scientific literature relative to the pathogen’s biology and disease’s pathobiology, more effective treatments, regarding the duration, are still current challenges [20]. Recent work has shown that the *Paracoccidioides* genus is composed of five species: *Paracoccidioides lutzii*, *Paracoccidioides restrepiensis*, *Paracoccidioides venezuelensis*, *Paracoccidioides americana*, and the cosmopolitan species *Paracoccidioides brasiliensis*, certainly the most investigated among the five species [21].

*Paracoccidioides* spp., like most living creatures, also need iron for the maintenance of cellular activity and consequently to obtain infectious success. This is evidenced in several ways. For example, the well-known MMcM (McVeigh and Morton) modified medium for growth of *Paracoccidioides* spp. contains in its composition a solution of trace elements, of which iron is the metal that appears in the highest molarity (approximately 3.6 µM; the next most abundant metal is zinc, with approximately 2.8 µM) [22]. Additionally, it has been shown that in the interaction with macrophages, the addition of iron chelators promotes a decrease in the pathogen’s survival, which points to the interdependence of the metal’s presence and successful parasitism [23,24].

In response to the iron deprivation imposed by the host, *Paracoccidioides* spp. modulate their metabolism prioritizing the use of glycolysis and negatively regulating iron-dependent pathways, such as the tricarboxylic acid cycle [25]. Additionally, the pathogen employs three high affinity and specific pathways for the uptake of iron, which allows the fungus to survive and persist, forming a ternary response [26,27,28]. Below we discuss these mechanisms and list new findings in this field of knowledge, ones that suggest this “little iron waltz” may mean improving therapeutic approaches to PCM.

## 2. “Time Signature”: The Transcriptional Reprogramming of *Paracoccidioides* spp. Facing Iron Deprivation

The ternary response of *Paracoccidioides* spp., when deprived of iron, resembles the rhythmic structure of a waltz, which features three beats per compass (ternary compass). Following this connotation, in this review we call a “beat” each of the three events that compose *Paracoccidioides* spp. response to iron deprivation. These events are orchestrated from a transcriptional reprogramming that occurs in the face of metal deprivation. Thus, still following the connotation of the performance of orchestrated music, the transcriptional reprogramming works just like the time signature at the beginning of a music sheet, which contains the global information for playing a song.

Iron deprivation suffered by pathogens when in the host leads to a transcriptional response that induces the expression of several proteins involved in obtaining metals from the host during infection. The expression of iron uptake pathways is part of a mechanism finely regulated by fungi, in which several transcriptional factors are activated to produce a rapid response to metal deprivation imposed by the host. As previously mentioned, the acquisition of iron from host by fungi is mediated by three uptake systems, of which one, the reductive iron assimilation pathway, is formed by ferric reductases [26]. The expression of these proteins is regulated by conditions such as metal deprivation and pH variation. In this way, the pathogenic fungus *C. albicans* has 15 putative ferric reductases, and the expression of some of those genes is under control of transcriptional factors; for example, ferric reductases FRE2p and FRP1p are regulated positively by the transcriptional factor Rim 101, in response to an alkaline environment [29]. In addition to the variation in response to pH, the expression of the ferric reductase FRP1p is increased during iron deprivation and the transcriptional factor Rim 101 also regulates the expression of this protein in *C. albicans* under this condition [30]. In *C. neoformans*, Rim 101 is involved in the use of heme from the host [31]. Additionally, the homeostasis of iron in *C. neoformans* is regulated by several other mechanisms. For example, the global repressor Tup1 is involved in the production of melanin; formation of capsules; and positive regulation of copper uptake transporter (CTR4p), ferric reductase transmembrane component (FRTp1), and siderophore–iron transporter (SIT2p) [32]. In *C. albicans*, Tup1 is also involved in the repression of genes such as *rbt5* for the capture of host heme [33,34]. In fungi of the *Paracoccidioides* complex, the expression of these transcriptional factors is still being elucidated. The expression of *Rim 101* and *Tup1* are induced during iron deprivation, and their expression seems to undergo a regulation mediated by miRNAs, differentially expressed during the deprivation of this metal (de Curcio, personal communication), indicating that these two genes are involved with the fine transcriptional response to iron deprivation, contributing to the control of iron homeostasis in fungi of this complex.

In *A. fumigatus*, the response to iron deprivation is regulated by transcription factors encoded by genes *sreA* and *hapX* [35,36]. The transcription factor SreA is reduced in iron deprivation, promoting the depression of HapX, an essential event for induction of the siderophore biosynthesis pathway [37]. In addition, the influence of SreA on genes involved in iron capture systems was observed [35]. Another gene that plays a role in the adaptation of *A. fumigatus* in low iron is *tptA*, which is homologous to the *Saccharomyces cerevisiae* gene *tpc1*. The study carried out by Huang and collaborators [38] showed that *tptA* loss reduces the expression of *hapX* and that the overexpression of *hapX* in the *tptA* mutant strain restored the growth defect and the production of siderophores. 

During iron deprivation, an increase in the expression of the transcriptional regulator *hapX* and in siderophores biosynthesis and uptake genes *sidA* and *sit1*, respectively, was observed in *P. brasiliensis* [25]. After 10 min of iron deprivation, the transcriptional regulator *hapX* level increased 2.3 times and remained increased up to 1 h. The *sidA* expression increased nine fold after 30 min of incubation and remained elevated for 24 h [25]. SrbA of *P. brasiliensis* restored the mutant’s defective growth phenotype of a null mutant strain of *A. nidulans*, demonstrating the functionality of this gene during iron deprivation [39]. As mentioned, mobilization of transcription factors orchestrates the response of fungi to Fe deprivation.

## 3. “The First Beat”: The Non-Classical Reductive Pathway of Iron Assimilation in *Paracoccidioides* spp.

The “first beat” of this “little iron waltz”, as mentioned above, is a strategy that some fungi use to circumvent iron deprivation, which is conventionally called the reductive pathway for iron assimilation and consists of multi-enzyme complexes that promote oxidation and reduction of the metal and internalize it [9]. Fungi such as *C. albicans* and *C. neoformans* have this ability, which is subsidized by iron reductases, copper-dependent ferroxidases and iron permeases that act together in the process [40,41]. This iron assimilation pathway allows the capture of free Fe^3+^, which is insoluble and therefore not bioavailable. However, since the amount of free metal in the host is minimal, this pathway is used mainly to sequester iron from host metalloproteins [40]. In *Histoplasma capsulatum*, a extracellular glutathione-dependent ferric reductase activity has been reported, in a process where glutathione is cleaved by a gamma-glutamyl transpeptidase (Ggt1) and a dipeptide with high reductive power is generated, acting on the reduction of Fe^3+^ that feeds the metal’s internalization system [42,43].

The investigation of the ability of *Paracoccidioides* spp. to use this reductive pathway began with in silico analyses that pointed to the presence in the fungus genome of ferric reductases (like *fre1*, *fre3*, *fre5*, *fre7*, and *frp1*), Ggt1, and copper-dependent ferroxidase orthologues (like *fetp*). However, it was surprising to find that *Paracoccidioides* spp. do not show ferric permeases [44]. As a way of investigating this path in more detail, a specific methodology was employed to accompany the steps of reduction, oxidation, and internalization of the metal in members of the genus *Paracoccidioides* [45]. This methodology, employing ^59^Fe uptake assays, suggested that the Fe uptake process occurs differently between *P. brasiliensis and P. lutzii*, so that *P. lutzii* has an inefficient reductive iron assimilation (RIA) pathway [26]. The absence of orthologs for ferric permeases also suggests that in the *Paracoccidioides* genus a non-classical reductive iron assimilation pathway (non-classical RIA) occurs [26]. The non-classical term refers to the fact that under iron deprivation, the transcripts of zinc permeases (Zrts) have positive regulation, suggesting that Zrts act in a promiscuous way, transporting both Zn^2+^ and Fe^2+^ during deprivation. In other words, Zrts would not be specific for zinc but a generalist divalent metal carrier [26]. In agreement with the in silico analyses that pointed to the presence of a Ggt1 orthologue in the genomes of *Paracoccidioides* spp., transcriptional data showed that Ggt1 undergoes positive regulation when the fungus is grown in iron deprivation [46]. However, the extent to which Ggt1 acts in conjunction with the non-classical RIA is still unknown. Once this question is resolved, new perspectives will certainly be outlined. This question has been the subject of studies by our group.

## 4. “The Second Beat”: Exploring the Host’s Iron-Containing Proteins

The “second beat” of this “waltz” is the ability of *Paracoccidioides* spp. to explore a host’s iron-containing proteins directly. In order to limit the toxicity of iron and the bioavailability of the metal for pathogens, the host associates the metal with various proteins, such as hemoglobin and myoglobin (in the form of a heme group), albumin, transferrin, ferritin, and lactoferrin, among others [47]. This variety of iron alternative sources caused pathogens to develop specific mechanisms for metal capture from those sources [9,40].

*C. albicans*, for example, uses a protein called Als3 to allow iron uptake from ferritin [40,48]. *C. albicans* is also able to use a host’s heme-containing proteins, especially hemoglobin and serum albumin, through a network of hemophores (Csa2, Rbt5, Pga7) located on the cell surface that allow the sequestration and internalization of these iron sources [28,33,49,50]. The performance of hemophores is based on the presence of the CFEM (common in several fungal extracellular membrane proteins) domain, which has been reported in other fungal species, such as *Candida glabrata* and *Candida parapsilosis* [51,52,53]. The internalization of the heme group by *C. albicans* is dependent on the ESCRT (endosomal sorting complex required for transport) endocytosis system [54]. Like *C. albicans*, *C. neoformans* also uses heme through the Cig1 hemophore, which differs from *Candida* spp. hemophores by not presenting the CFEM domain [55]. The internalization of heme by *C. neoformans* is also dependent on the ESCRT system [56,57].

*Paracoccidioides* spp. are capable of using various iron-containing proteins such as lactoferrin, ferritin, transferrin, and hemoglobin [27]. Regarding hemoglobin uptake, the event occurs through a receptor that also has the CFEM domain, homologous to *C. albicans* Rbt5. The knock-down of *rbt5* compromises the capacity of *Paracoccidioides* spp. survive on macrophages and colonize murine’s spleen [27]. Despite these data, it remains to be clarified whether *Paracoccidioides* spp. use only Rbt5 for using heme/hemoglobin or if other hemophores and ancillary proteins may be involved. It is important to highlight that in different *Paracoccidioides* genomes there are at least four sequences that code for proteins that contain the CFEM domain (Table 1). There is still a need for elucidating whether those proteins are involved in the capture of iron-containing proteins from the host. Additionally, it is not clear yet which endocytic system the fungus uses to internalize the molecule. It is also necessary to investigate whether *Paracoccidioides* spp. have other proteins dedicated to the use of specific iron sources in the host.

## 5. “The Third Beat”: Biosynthesis and Uptake of Siderophores by *Paracoccidioides* spp.

Finally, but just as important as the other events described so far, there is the “waltz’ third beat”: the non-reducing mechanism of iron uptake mediated by siderophores [58]. From the Greek “iron carriers”, siderophores are low-molecular-weight (usually 1 KDa) molecules that solubilize ferric iron (Fe^3+^). Siderophores can also act in the storage of iron [59]. The iron uptake process by this mechanism occurs by excretion of siderophores and binding to the free iron ions, forming the iron–siderophore complex. After uptake, this complex is linked to specific receptor proteins, like Sit1 of *C. albicans* and *C. neoformans* and, MirB and MirC of *Aspergillus nidulans*, present in the cell membrane and through active transport are internalized [60,61,62]. There are different types of siderophores. Carboxylate siderophores are produced mainly by bacteria such as *Mycobacterium tuberculosis*, which synthesizes carboxymicobactin [63]. Catecholate siderophores are produced mainly by bacteria, like enterobactin (or enterochelin) produced by *Escherichia coli* and other Enterobacteriaceae [64]. However, most of the siderophores produced by fungi belong to the hydroxamate group, with the exception of the rizzoferrin, siderophores of the carboxylate type, which are produced by several *Mucorales* sp., as well as the catecholate-type pistilarin, produced by *Penicillium bilaii* [17].

Hydroxamate siderophores are the most commonly found in nature and are grouped into four structural families: rhodotoluric acid, fusarinines, coprogens, and ferricromes [17]. In bacteria, hydroxamates are composed of acylated and hydroxylated alkylamines, while in fungi they are derived from ornithine, a non-proteinogenic amino acid, which is hydroxylated and alkylated [65]. The importance of siderophores in fungal virulence is well characterized by detailed studies with the pathogen *A. fumigatus*, which produces siderophores belonging to the ferricrome family (ferricrocin), which are intracellular, and fusarinins (fusarinin C, triacetylfusarinica (TAFC)), which are extracellular. The production of intracellular ferricrocin siderophores is coordinated according to the morphology of the fungus; that is, ferricrocin is produced during filamentous growth, while hydroxyferricrocin is produced in spores of conidia, which are the infectious particles [9]. The iron–siderophore complex is transported back to cells via membrane proteins called siderophore transporters. In *A. nidulans*, MirA, MirB, and MirC were characterized as transporters of siderophores, with MirA and MirB being carriers of enterobactin and TAFC, respectively. Although MirC shows a high degree of conservation with other siderophore transporters, its exact role in capturing siderophores has not yet been found [66].

The mechanisms of iron uptake mediated by siderophores in fungi of the *Paracoccidioides* genus were investigated. These fungi conserve homologues genes related to the biosynthesis and uptake of siderophores, and transcripts of such genes are positively regulated during iron deprivation [44,46]. The genomes of members of the *Paracoccidioides* genus encode the orthologue genes for siderophore biosynthesis, namely sidA, sidF, sidC, sidD, sidI, and sidH, as well as the orthologues for the capture of siderophores, namely sit1, mirB, and mirC. During iron deprivation in *Paracoccidioides* spp., the synthesis and secretion of siderophores of the hydroxamate type coprogen B and dimerumic acid occurs, which are extracellular siderophores, and ferricrocin and ferricromes C, which are intracellular siderophores. The capacity for synthesis and secretion of siderophores by *Paracoccidioides* spp. was attested by a crossfeeding experiment, whereby a strain of *A. nidulans* unable to produce siderophores had its growth restored when grown in MMcM without iron, in co-cultivation with *Paracoccidioides* spp., which pointed out that *A. nidulans* can use siderophores produced and secreted by *P. brasiliensis* [28].

Members of the *Paracoccidioides* genus can use xenosiderophores as a source of iron, such as ferrioxamine B (FOB) [67,68]. In the presence of FOB, the suppression on synthesis of endogenous siderophores occurs, indicating the use of ferric ions linked to FOB as an iron source. Additionally, proteomic and enzymatic analyzes demonstrated that SidA, the enzyme that promotes ornithine hydroxylation during siderophore biosynthesis, was negatively regulated, suggesting the blockade of the pathway due to the alternative source of iron. To investigate the role of siderophore production in fungus virulence, *P. brasiliensis* knockdown strains for *sidA* were obtained, depicting a reduction in the production of siderophores as well as a reduction in the pathogenicity in the *Tenebrio molitor* model of infection, when compared to wild type strains [68].

As mentioned above, members of the *Paracoccicioides* genus can use xenosiderophores [67,68]. Since FOB is a hydroxamate siderophore, a class produced by the genus, it seems interesting to question whether *Paracoccidioides* spp., such as *A. nidulans*, would have the ability to transport siderophores from groups other than hydroxamates, produced by bacteria for example. In this sense, with the intention of pointing out aspects of the response of *Paracoccidioides* spp. to iron deprivation that need to be clarified, we performed molecular dynamics simulation. The three-dimensional structures of the MirB, MirC, and Sit1 proteins of *Paracoccidioides* spp. were subjected to molecular modeling using the I-TASSER server [69]. The structures of the siderophores enterobactin, ferrioxamine B, and carboxymicobactin were obtained in the literature [70,71,72], and molecular docking was performed between siderophores and proteins using AutoDock Vina [73]. Based on the quality of obtained data (Appendix A), we present here preliminary results based on molecular dynamics of the *P. brasiliensis* siderophore receptors suggesting capture of hydroxamates, catecholates, or carboxylates, as shown in Figure 1. The energy interaction data are presented in Appendix A and suggest that although all *P. brasiliensis* siderophores receptors have the apparent ability to interact with any type of siderophore (for additional information on molecular dynamics experiments, see Appendix A), MirB would have a higher affinity for carboxymycobactin (a carboxylate, Figure 1A), and MirC would have a higher affinity for enterobactin (a catecholate, Figure 1B), while Sit1 may have a higher affinity for FOB (a hydroxamate, Figure 1C). In accordance with the molecular dynamics data, we carried out analyzes of the transcripts of the siderophore transporters *sit1*, *mirB*, and *mirC.* For this, *P. brasilienses* yeast cells were incubated in MMcM supplemented with BPS or FOB, and after 6 and 24 h, the cells were collected and extraction of total RNA was performed. Transcriptional analyzes were performed using the standard curve method for relative quantification [74] for calculating the relative expression levels of transcripts of interest. Surprisingly, the transcripts of *sit1*, *mirB*, and *mirC* increased in the first 6 h in the presence of FOB. However after 24 h, it was noticed that the transcript of *sit1* remained elevated, corroborating the possible preferential interaction of Sit1 with FOB (Figure 1D). Obviously, these data are preliminary and require further investigation, but have information that points to the capacity of *Paracoccidioides* spp. of using any type of siderophore, which is exciting because it could yield discoveries about the biology of the pathogen not explored to date.

## 6. “The Addition of Chialteras”: Recent Findings and Future Perspectives on the Response of *Paracoccidioides* spp. to Fe Deprivation

In this review, we compare the response of *Paracoccidioides* spp. to iron deprivation and the execution of a waltz. In music, notes (like chialteras) can suddenly appear and surprise those who hear. Similarly, in science, new discoveries can bring new concepts and break paradigms. It is justified to continue studying the mechanisms that *Paracoccidioides* spp. employ when challenged by iron deprivation, as shown in Figure 2. We also cite some events that occur in other fungal species when subjected to iron deprivation, as summarized in Table 2.

Our group has been working on this challenge, and soon new information about proteins of the pathogen involved in the capture of iron from the host and even the participation of miRNAs in this process will be published. In this review, we also present preliminary data that point out that *Paracoccidioides* spp. is capable of using siderophores of all classes, including carboxymycobactin, a siderophore produced by *M. tuberculosis*, which is a bacterium that has already been described as causing coinfection with *Paracoccidioides* spp. [75,76]. This is a very relevant fact to be studied, considering that in a recent work by our group, it was seen that offering the FOB xenosiderophore to *P. brasiliensis* causes downregulation of SidA, the initial enzyme in the siderophore biosynthesis pathway, which raises questions about how pathogens respond to the presence of xenosiderophores in vivo [68]. In this sense, characterizing the potential xenosiderophores used by the *Paracoccidioides* complex aggregates information about the mechanisms used by these fungi for the development in conditions of iron deprivation, such as that found during the colonization of the host. In addition, all this data point to the challenge of understanding not only how *Paracoccidioides* spp. interact with the host but also with other pathogenic microorganisms and even the commensal microbiota. Certainly, all these findings will add more and more compasses to this “waltz” and, from a “little iron waltz”, soon we will be able to appreciate a “complete symphony”.

## Figures and Tables

**Figure 1 jof-06-00221-f001:**
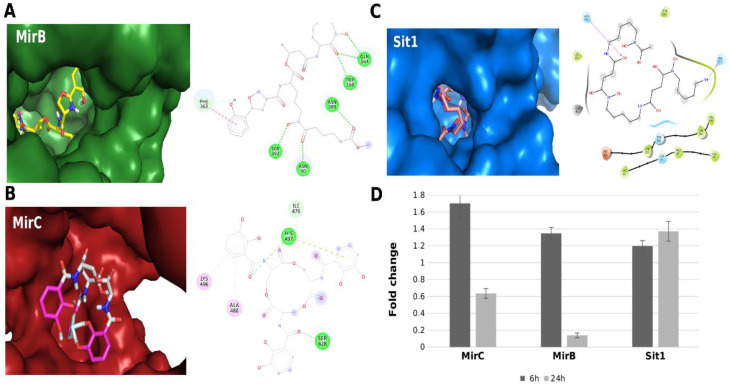
Interactions between three-dimensional models and siderophores. (**A**) Interaction between MirB and carboxymycobactin. The left panel shows the coupling of siderophore in the MirB pocket (green). In the right panel, it is observed that the interaction is maintained by conventional hydrogen bonds (green), carbon–hydrogen bonds (light blue), and pi–pi stacked interaction (pink) between the aromatic rings of carboxymycobactin and PHE362. (**B**) Interaction between MirC and enterobactin. The left panel shows the coupling of the siderophore in the MirC pocket (red). In the right panel, the interaction is maintained by conventional hydrogen (green) and carbon–hydrogen (light blue) bonds, in addition to the pi–alkyl (pink) interaction between the aromatic enterobactin ring and the amino acids LYS496 and ALA480 and the pi–cation interaction (yellow) between the aromatic ring and the HIS497. (**C**) Interaction between Sit1 and ferrioxamine B. The left panel shows the coupling of the siderophore in the Sit1 pocket (blue). In the right panel, it is observed that the ligand ferrioxamine B is maintained mainly by hydrophobic affinity (green) to the pocket of Sit1. (**D**) Levels of transcripts from siderophore transporter genes accessed by RT-qPCR after cultivation of *P. brasiliensis* in Fe deprivation or in the presence of FOB for 6 h and 24 h. For a list of the primers used in the experiments, see Appendix A. Additional information on this transcriptional analysis is available in the Appendix A.

**Figure 2 jof-06-00221-f002:**
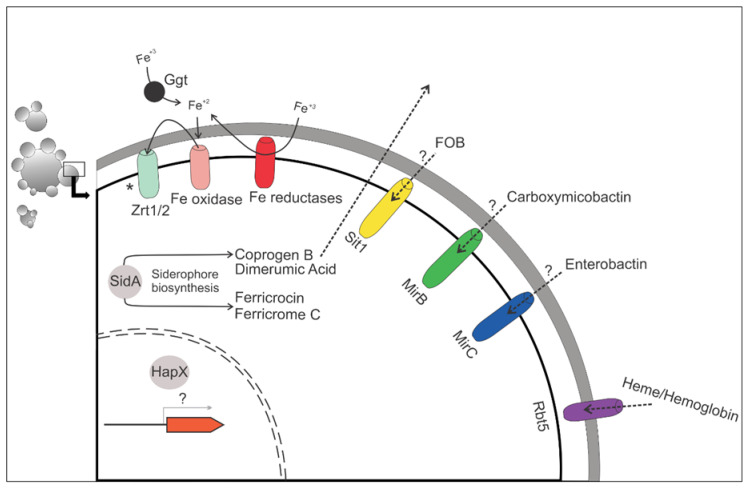
Overview of the mechanisms of iron uptake used in species of the *Paracoccidioides* complex. During iron deprivation *Paracoccidioides* spp. employ several mechanisms of iron uptake, such as siderophore transporters (Sit1, MirB, and MirC); binding to the heme group, mediated by Rbt5; or by iron and zinc transporters and iron-reductases. Rbt5 and SidA are described as virulence factors in *Paracoccidiodes* spp. “*” In *P. lutzii*, RIA is apparently ineffective, because after the metal reduction step, there is no uptake of this. “?” points to aspects that still require elucidation.

**Table 1 jof-06-00221-t001:** The putative CFEM proteins in *Paracoccidioides* spp. Genomes *.

Species ^a^	Gene ID ^b^	Product Description ^c^	CFEM (E-Value) ^d^	GPI Modification Site Prediction? ^e^	SignalP ^f^	SecretomeP ^g^
*P. lutzii*	PAAG_04763	Hypothetical protein	2.0 × 10^−8^	None	Yes	-
	PAAG_11627	Hypothetical protein	2.8 × 10^−13^	None	Yes	-
	PAAG_05158	Rbt5 ^#^	3.3 × 10^−15^	Yes	Yes	-
	PAAG_02225	Csa1 ^#^	5.2 × 10^−12^	None	Yes	-
	PAAG_00918	Hypothetical protein	5.5 × 10^−11^	None	Yes	-
*P. brasiliensis*	PADG_11659	Hypothetical protein	1.5 × 10^−11^	None	-	-
	PADG_05363	Csa1 ^#^	1.5 × 10^−11^	None	Yes	-
	PADG_02506	Hypothetical protein	1.7 × 10^−8^	None	Yes	-
	PADG_03909	Hypothetical protein	2.1 × 10^−8^	Yes	Yes	-
	PADG_06374	Hypothetical protein	3.6 × 10^−13^	None	-	Yes
	PADG_05000	Rbt5 ^#^	4.5 × 10^−15^	Yes	Yes	-
*P. americana*	PABG_12009	Hypothetical protein	1.6 × 10^−11^	None	Yes	-
	PABG_00115	Hypothetical protein	1.7 × 10^−8^	None	Yes	-
	PABG_01323	Hypothetical protein	2.4 × 10^−8^	Yes	Yes	-
	PABG_04599	Rbt51 ^#^	4.5 × 10^−15^	Yes	Yes	-

* The search for probable CFEM proteins in the genomes of *Paracoccidioides* was carried out using the online tool available at https://fungidb.org/fungidb/ (genes; sequence analysis; protein motif pattern). The term used in the search was CFEM. ^a^
*Paracoccidioides* species with genomes available in the FungiDB database. ^b^ Access code of the predicted CFEM-containing sequences. ^c^ Annotation available on FungiDB. ^d^ Confidence score of the prediction for CFEM domain presence. ^e^ Site prediction for GPI anchor modification. The prediction was performed by the online tool big-PI Predictor available at http://mendel.imp.ac.at/gpi/gpi_server.html. ^f^ Classical secretion prediction performed by the online tool SignalP 4.0 available at http://www.cbs.dtu.dk/services/SignalP-4.0/. ^g^ Non-classical secretion prediction performed by the online tool SecretomeP 2.0 available at http://www.cbs.dtu.dk/services/SecretomeP/. ^#^ Names assigned to the sequences as suggested by Bailão et al. (2014) [27].

**Table 2 jof-06-00221-t002:** Mechanisms of fungi response to iron deprivation.

Species	Described Mechanism	Proteins Involved
*Paracoccidioides* spp.	Non-classical RIA ^a,b^	Ferric reductases, ferroxidase, GGT, Zrt1/2
	Use of host Fe-proteins	Rbt5 (heme/hemoglobin uptake)
	Biosynthesis and uptake of siderophores	SidA, Sit1, MirB, MirC
*Candida albicans*	RIA	Ferric reductases, ferroxidases, ferric permeases
	Use of host Fe-proteins	Rbt5, Pga7, Csa2 (heme/hemoglobin uptake); Als3 (iron uptake from ferritin)
	Uptake of siderophores	Sit1
*Cryptococcus neoformans*	RIA	Ferric reductases, ferroxidases, ferric permeases
	Use of host Fe-proteins	Cig1 (heme/hemoglobin uptake)
	Uptake of siderophores	Sit1
*Aspergillus* spp.	Biosynthesis and uptake of siderophores	SidA, MirA, MirB, MirC

^a^ RIA: reductive iron assimilation. ^b^ Non-classical RIA: in *Paracoccidioides* spp. RIA is referred to as non-classical because the internalization of iron is not performed by a ferric permease but by a zinc/iron permeases called Zrt1/2.

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
