# Peer review of "The “Little Iron Waltz”: The Ternary Response of Paracoccidioides spp. to Iron Deprivation"

_jof, 2020, doi:10.3390/jof6040221_

Round 1
Reviewer 1 Report
This is a nicely written thorough review on the Paracoccidioides response to iron deprivation seen as a three-steps organised waltz. After reading it these are my suggestions to the authors:
- I consider this review would benefit from a figure illustrating the 3 beats of the Paracoccidioides waltz, including differences among species, the new data provided by the authors in the review and areas that still need further study and description.
- Also, I would add a Table summarising each beat in other fungal species (that are described in the text like C. albicans or C. neoformans) compared to Paracoccidioides spp.
- Sentences in lines 117 to 120 need references
- Sentence from204 to 205, what fungal species you are referring to? or are these receptors universal to all fungi?
- Line 240, references have a different format
- A brief summary of the molecular dynamics and gene expression methodology need to be provided in the main text and not only making a reference to supplementary material for more info. Also, authors need to highlight the aim of performing these experiments. This section of the review is interesting but it is lacking the otherwise nice flow that the rest of the review has.
Reviewer 2 Report
The review is well organized and provides an overview on the mechanisms used by Paracoccidioides to acquire iron from the host. A deep knowledge of the mechanisms used by these fungi for capturing iron from the host is the base for the development of possible treatments, and for these reasons the review is of wide interest for readers.
My main observation is releated to the last part of the review, that is not a review but a research paper, and my suggestion is to leave this part for a pure experimental paper, in particular because it is just related to one of the three mechanism of iron capture.
Minor points:
row 97: remove comma between "101" and "also"
chapter 5: there quite a large part describing siderophores in bacteria. For simmetry, I would expand comparison with bacteria also in the other two mechanism, or reduce here.
row 238: add a periods at the end of the sentence after "investigation".
Author Response
Response to Reviewer 2 Comments
Point 1: My main observation is related to the last part of the review, that is not a review but
a research paper, and my suggestion is to leave this part for a pure experimental paper, in
particular because it is just related to one of the three mechanism of iron capture.
Response 1: Thank you very much for your observation. We agree that they are new data of a
preliminary nature. However, with the intention of pointing out aspects of the response of
Paracoccidioides spp. that need to be clarified, we would like to keep this part in the review.
We believe that the publication of this preliminary data in this review may stimulate further
investigations involving this context.
Point 2: row 97: remove comma between "101" and "also".
Response 2: Thanks for the correction. The comma has been removed.
Point 3: chapter 5: there quite a large part describing siderophores in bacteria. For symmetry,
I would expand comparison with bacteria also in the other two mechanism, or reduce here.
Response 3: We thank you for this suggestion. We choose to reduce the part that referred to
bacterial siderophores.
Point 4: row 238: add a period at the end of the sentence after "investigation".
Response 4: Thank you for this correction. A period has been added as requested and now it
is on line 229.